# Dual-Shell Microcapsules for High-Response Efficiency Self-Healing of Multi-Scale Damage in Waterborne Polymer–Cement Coatings

**DOI:** 10.3390/polym16010105

**Published:** 2023-12-29

**Authors:** Chenyang Liu, Zhicheng Sun, Shouzheng Jiao, Ting Wang, Yibin Liu, Xianyu Meng, Binbin Zhang, Lu Han, Ruping Liu, Yuanyuan Liu, Yang Zhou

**Affiliations:** 1Beijing Engineering Research Center of Printed Electronics, Beijing Institute of Graphic Communication, Beijing 102600, China; liuchenyanglh@163.com (C.L.); m18435764908@163.com (T.W.); mrbin970811@gmail.com (Y.L.); hanlu@iccas.ac.cn (L.H.); liuruping@bigc.edu.cn (R.L.); 2School of Chemistry and Chemical Engineering, Harbin Institute of Technology, Harbin 150090, China; meetjiao@gmail.com; 3State Key Laboratory of Special Functional Waterproof Materials, Beijing Oriental Yuhong Waterproof Technology Co., Ltd., Beijing 100123, China; mengxy01@yuhong.com.cn (X.M.); zhangbb03@yuhong.com.cn (B.Z.); 4Key Laboratory of Advanced Materials of Tropical Island Resources, Ministry of Education, School of Chemistry and Chemical Engineering, Hainan University, Haikou 570228, China; yyliu@hainanu.edu.cn

**Keywords:** microcapsules, self-healing, waterborne polymer–cement, coatings, silica gel shell

## Abstract

Waterborne polymer–cement coatings have been widely applied in building materials due to their organic solvent-free nature, low cost, and eco-friendliness. However, these coatings can easily crack during the drying process as a result of construction environment factors, compromising the barrier performance of the coating and limiting its large-scale application. In this study, a dual-shell self-healing microcapsule was developed, which can effectively heal damage on a macro scale in waterborne polymer–cement coatings. Specifically, this dual-shell self-healing microcapsule was designed with a silica gel shell and a tannic acid–cuprum (TA–Cu) double-shell structure embedded with an epoxy resin (EP) healing agent, which was successfully fabricated via a two-step in situ polymerization. This silica gel shell self-healing microcapsules can effectively load into waterborne polymer–cement coatings. As the coating dries and solidifies, the silica gel shell of the microcapsule also becomes loose and brittle due to dehydration. This improves the mechanical initiation efficiency of the microcapsules in the coating. This study provides a novel approach for the application of self-healing microcapsules in waterborne coating systems, which can significantly reduce cracking during the drying process of waterborne polymer–cement coatings and improve the service life of the coating under complex conditions.

## 1. Introduction

Waterborne polymer–cement coatings have been widely used in indoor and outdoor building facilities due to their green, environmentally friendly, and good barrier features [1,2,3]. However, waterborne coatings can be easily affected by various environmental factors such as temperature and humidity during use [4]. During the drying process of the coating, different drying rates in different areas may lead to local volume shrinkage and serious cracking. The resulting cracked coating directly loses its protective function, which needs to be completely scraped off and recoated. Moreover, reconstruction leads to the production of a large amount of coating waste [5]. To achieve the sustainable development of construction coating, therefore, it is essential to develop approaches to effectively reduce the cracking of these waterborne coatings.

Self-healing microcapsules are a common type of extrinsic self-healing material [6]. Under the external forces and cracks, the microcapsules loaded with liquid healing agents would crack at the same time to release healing agents in situ, which can effectively fill and solidify the cracks under the dual effects of their fluidity. With the capillary phenomenon in the cracks, self-healing of the damage can be achieved [7,8,9]. Compared with intrinsic self-healing materials based on reversible bonds, this self-healing system can effectively fix large-scale crack damage. The corresponding self-healing materials are easy to use and flexible, and their preparation cost is low, so they have been widely used in various fields such as concrete [10], functional coatings [11,12], and electronic circuits [13]. To date, the application of self-healing microcapsules in the field of waterborne coatings remains limited [14,15]. The main issue lies in that the stress generated inside the waterborne coating is greatly low during the drying process, so is not strong enough to cause the cracking of the microcapsules in the coating [16,17]. So, these current microcapsules directly lose crack ability if they are present in a coating. Microcapsules with weaker structural strength may be damaged during the storage and stirring of the coating, which can effectively trigger the self-healing performance under low stress [8,18]. Therefore, it is of great interest to develop self-healing microcapsules with thin walls. 

Some researchers have employed nanoparticle-dispersed healing agents to prepare Pickering emulsions, which have been utilized as thin-shell microcapsules applied in waterborne coatings [19,20]. For example, Xiao et al. reported the encapsulation of GMA with graphene oxide to produce a high-loading-capacity thin-shell microcapsule (GMA@GOMC) [21]. They then embedded the microcapsules in a waterborne EP matrix to obtain a self-healing coating on the surface of a hot-dip galvanized steel plate. Zhang et al. employed a linseed oil (LO) Pickering emulsion stabilized using nanocellulose (CNC) as a thin-shell microcapsule to prepare a highly transparent self-healing waterborne polyurethane (WPU) coating with a healing efficiency of up to 65% [22]. However, thin-shell microcapsules based on Pickering emulsions have issues, including weak structural stability and poor sealing, leading to ease of being damaged during the coating preparation [23,24,25]. In addition, they are subject to the effects of catalysts in the coating, causing the embedded healing agent to solidify prematurely and making it difficult to store in the coating for a long time. Therefore, it remains a challenge to prepare a self-healing microcapsule that can function in waterborne coatings during a long storage period.

The sol–gel method is a conventional approach for preparing silica, with ethyl silicate used as the silicon source, which undergoes hydrolysis in alcoholic solution under acidic conditions. By adjusting the pH to alkaline, condensation reactions occur between silanol molecules [26,27]. As the condensation progresses, the polymers in the sol increase in size, forming a three-dimensional network structure that encapsulates the solvent within its pores, resulting in a silica gel shell. This silica gel has a loose structure upon dehydration, making it sensitive to drying. Tannic acid, a water-soluble natural polyphenolic compound, undergoes chelation reactions with copper ions under alkaline conditions through its catechol groups. Metal–phenolic networks are commonly used as shell or coating agents to enhance the sealability of microcapsules [28], offering simplicity in preparation and environmental friendliness. Double-shelled microcapsules of silica gel combined with TA-Cu have a very wide range of applications in the field of stimulus response and are a very valuable potential solution for the preparation of self-healing microcapsules. However, there are no studies that have validated the use of such drying-sensitive microcapsules in water-based polymer coatings.

In this study, a self-healing microcapsule with a silica gel shell embedded with EP and a double shell of TA-Cu was designed through an in situ polymerization method. The silica gel wall material of the microcapsule remained stable under vigorous stirring conditions. Moreover, the metal–phenolic network formed via the chelation reaction of TA and Cu^2+^ was deposited on the surface of the EP@SiO_2_ microcapsules. The fabrication of dual walls not only increased the strength of the microcapsule structure but also filled the defective micropores on the silica gel. In this case, the healing agent was not prematurely triggered by the curing agent, and the microcapsule could also be effectively loaded in the waterborne polymer cement coating. After the drying of the coating, the silica gel shell of the microcapsule also dried, which became loose and easily broken. The resulting shell could readily crack and heal with a more sensitive response. We presume that this solution might address the problem of poor compatibility between traditional self-healing microcapsules and layers to effectively improve the self-healing efficiency of self-healing microcapsule systems in aqueous coatings. Specifically, the tensile strength and healing efficiency of coatings under different additive amounts were systematically explored. In addition, the healing efficiency of internal microcracks was also investigated. The coating could effectively heal macro-scale scratch damage, and the crack performance and healing efficiency were studied. We anticipate that this novel approach for the application of self-healing microcapsules in waterborne polymer–cement coatings systems has great potential for practical application in the field of waterborne coatings.

## 2. Materials and Methods

### 2.1. Materials

Tetraethyl orthosilicate ester (TEOS) and ammonium hydroxide were supplied by Shanghai Aladdin Biochemical Technology Co., Ltd. (Shanghai, China). Arabic gum (AG), tannic acid (TA), copper chloride, hydrochloric acid, and ethanol were provided by Beijing Innochem Science & Technology Co. (Beijing, China). Epoxy resin (EP) and curing agent (AT30) were purchased from Easy composites (Beijing, China) Technology Co., Ltd. (Beijing, China). Water polymer cement coatings were mainly composed of vinyl acetate-ethylene copolymer (VAE) water-based emulsions and silicate cement mixtures, which were generously provided by the State Key Laboratory of Special Functional Waterproof Materials (Beijing, China).

### 2.2. Preparation of EP@SiO_2_@TA-Cu Microcapsules

A mixture of TEOS (10 g) and ethanol (10 g) was created in a beaker with deionized water (20.0 g). After the pH was adjusted to 2–3 using hydrochloric acid, the mixture was heated at 45 °C for 30 min to yield a silica sol solution. In addition, 3 g of Arabic gum (AG) was used as a surfactant. It was dissolved in 100 g of deionized water. Under a water bath at 50 °C, epoxy resin (EP) was gradually added to the AG solution. The mixture was then emulsified for 1 h at a stirring speed of 700 rpm, resulting in a stable EP emulsion. Then, the silica sol solution was added to the emulsion solution, whose pH was adjusted to around 8.0 using the ammonia solution (2 mol/L). After stirring for 8 h, the initial silica gel shell microcapsules were obtained. Following the method proposed by Cao et al. [28], TA (1.0 g) was fully mixed with the above suspension, and copper chloride (1.0 g) was added dropwise under stirring conditions to form EP@SiO_2_@TA-Cu microcapsules.

### 2.3. Preparation of Polymer Emulsion Cement Waterproof Coating Containing Microcapsules

The microcapsules and curing agent (AT30) were slowly added to the coatings under magnetic stirring for 15 min. The coating samples were then placed in a low-pressure oven for about 10 min to eliminate trapped air bubbles. The coating samples were then applied on polytetrafluoroethylene sheets with a wet film thickness of 2 mm using a doctor-blade film applicator. The specimens were then cured for about 72 h at room temperature. The thickness of the cured coating samples was measured to be 1.7 ± 0.1 mm.

### 2.4. Characterization 

Sufficient images of the microcapsules were obtained via scanning transmission electron microscopy (SEM) (Gemini 300, ZEISS, Jena, Germany) measurements in electron-dispersive spectroscopy (EDS) mode. The average particle size and size distribution of prepared microcapsules were determined using a Malvern Mastersizer 2000 (UK) via particle size analysis. Fourier-transform infrared intensity (FTIR) spectroscopy measurements were performed with a FTIR spectrometer (Nicolet iS20, Thermo Scientific, Waltham, MA, USA) to determine the functional groups. A monochromatic Al K alpha source with a 400 μm beam diameter was used for the XPS analysis (K-Alpha, Thermo Scientific, Waltham, MA, USA). A simultaneous thermal analyzer (STA 449 F5 Jupiter, Netzsch, Germany) was employed to test the microcapsule’s thermal stability. 

The tensile performance of the coating was tested according to the tensile test method in ASTMD412 using a universal testing machine (CMT6103, MTS, Minnesota, MN, USA) [29]. By cutting the coatings into standard specimens using a cutter and stretching at a rate of 200 mm/min, the tensile strength (R_m_) of self-healing coatings with intact coatings, damaged coatings, and after-healing coatings was tested. The self-healing efficiency of microcracks (HE_mc_) was calculated using Equation (1).
(1)$${HE}_{mc}=\frac{R_{m}(Healing)-R_{m}(Damaged)}{R_{m}(Intact)-R_{m}(Damaged)}\times 100\%$$

The surface topography and depth (D) of the coating cracks were tested using laser confocal microscopy [30]. A cut was made on the coated sample with a blade at the crack location, and the surface topography and depth of the coating before and after healing were scanned. The filling state of the healing agent on the crack during the healing process was observed. The filling rate (HE_fr_) was calculated using Equation (2).
(2)$${HE}_{fr}=\frac{D(After\;healing)}{D(Before\;healing)}\times 100\%$$

The self-healing coating was coated on an iron sheet and dried [31], and 1 cm long damage was made on the coating with a blade. The damaged coating was placed at room temperature for 24 h to analyze the coating after healing. Scratched coatings were immersed in 3.5 wt% NaCl solution for 24 h. A conventional three-electrode system was employed (consisting of working electrode, saturated calomel electrode, and platinum sheet electrode). The area of the working electrode tested via EIS was 3.14 cm^2^. EIS was performed in the frequency range from 10^5^ Hz to 0.01 Hz using an alternating current signal amplitude of 10 mV. The self-healing efficiency of barrier performance (HE_bp_) was calculated using Equation (3).
(3)$${HE}_{bp}=\frac{Log({\left|Z\right|}_{0.01,Healing})-Log({\left|Z\right|}_{0.01,Damaged})}{Log({\left|Z\right|}_{0.01,\mathrm{Intact}})-Log({\left|Z\right|}_{0.01,Damaged})}\times 100\%$$

## 3. Results and Discussion

### 3.1. Preparation and Characterization of Self-healing Microcapsules

The synthesis scheme of the self-healing microcapsules is presented in Figure 1. This microcapsule with a silica gel shell was prepared via in situ polymerization. By adding EP to a small amount of butyl glycidyl ether (BGE) and mixing evenly, an oil phase was obtained. The addition of BGE greatly enhanced the fluidity of the healing agent, facilitating its flow and filling of cracks via capillary action. The emulsion was stabilized through the emulsifier Arabic gum and the positively charged cetyltrimethylammonium bromide, which existed as a dispersed phase in the continuous phase water at the interface of the two phases. Specifically, TEOS was used as a silicon source within an ethanol solution and hydrolyzed into a silanol solution under acidic conditions. When the silanol solution was dropped into the emulsion, the silanol accumulated and cross-linked on the surface of the emulsion droplets under the attraction of a positive charge, forming a uniformly textured silica gel shell; the reaction equation is shown in Appendix A [32]. To further enhance the strength and seal of the microcapsules, a chelation reaction between equimolar TA and Cu^2+^ was conducted under weakly alkaline conditions to form a metal–phenol network on the surface of the EP@SiO_2_ microcapsules (the structural formula is shown in Appendix A), thus creating structurally stable dual-shell capsules, EP@SiO_2_@TA-Cu [33]. The microcapsules could be directly added to water-based coatings in their wet state. During the drying process of the coating, the silica gel shell of the microcapsules lost moisture and became loosely structured silica dioxide. Therefore, it was more susceptible to small forces that trigger rupture and thus achieve higher repair efficiency.

The spherical shape, size, and integrity of the double-shell structure of the microcapsules were observed using scanning electron microscopy (SEM). Figure 2a,b show the microcapsules in a uniform spherical shape. A thin film formed using TA-Cu chelate compounds was deposited on the surface of the microcapsules. Microcapsule rupture, induced by an external force, is shown in Figure 2c, and the wall thickness of the microcapsules was approximately 2–3 μm. Due to the mechanical stirring in the emulsification process of microcapsule preparation, the liquid velocity varied between the vicinity of the propeller and the edges of the container, leading to a normal distribution of microcapsule sizes, with an average diameter of 64 μm (Appendix A). In Figure 2d, the SEM-EDS element mapping shows a uniform distribution of C, O, Si, and Cu on the surface of the EP@SiO_2_@TA-Cu microcapsules, demonstrating the even distribution of Cu elements and the formation of metal–phenolic networks via TA and copper ions on the microcapsule surface. Fourier-transform infrared spectroscopy (FTIR) was used to identify the chemical bonds and functional groups of the microcapsules. As shown in Figure 2e, the absorption bands at 917 cm^−1^ and 831 cm^−1^ were attributed to the characteristic absorption peaks of the epoxy groups in EP, proving that the healing agent EP was successfully encapsulated. The band at 1100 cm^−1^ was attributed to Si–O–Si stretching vibrations in the silica wall material, mainly originating from the silica gel shell. The absorption peak near 1711 cm^−1^ was due to the –C=O stretching vibration on tannic acid, confirming the successful modification of the microcapsule surface via the metal–phenolic network formed via the chelation reaction of TA with copper ions. Additionally, X-ray photoelectron spectroscopy (XPS) was used to further analyze the surface chemical properties of the microcapsules. Figure 2f presents XPS images of EP, EP@SiO_2_, and EP@SiO_2_@TA-Cu. As compared to EP, EP@SiO_2_ microcapsules showed the presence of Si elements, with a content of about 3.2%, while the dual-shell EP@SiO_2_@TA-Cu microcapsules showed the presence of copper absorption, at about 1.3%. As shown in Appendix A, the high-resolution C1s peak deconvoluted to 284.8, 286.7, 286, and 289 eV, corresponding to C–C, C–O–Ph, C–O–C/C–OH, and C=O functionalities in TA, respectively. Similarly, the O1s spectrum deconvoluted to 533.0 and 532.4 eV, representing the organic C–O and O–H structures, respectively. The absorption peaks of Si2p at 102.3 and 103.5 derived from C-Si-O and SiO_2_, respectively, further confirmed the successful encapsulation of EP@SiO_2_@TA-Cu bilayer shell in the self-healing microcapsules [34]. Thermogravimetric analysis revealed the thermal stability of the microcapsules, as presented in Figure 2g. The thermal decomposition onset temperature of EP was approximately 179 °C, which increased to 269 °C after encapsulation with silica gel, suggesting the protective effects from encapsulation and the enhanced thermal stability of EP. After the formation of the second shell via TA-Cu chelation, the initial thermal decomposition temperature of the microcapsules dropped to 229 °C. This decrease was primarily due to the relatively poor thermal stability of the TA-Cu shell, which initially decomposed during the heating. These data collectively proved the presence of elements and compounds unique to the EP, silica gel shell, and metal–phenolic network, thereby confirming the successful preparation of the microcapsules. The microcapsules’ structural stability was tested by calculating the morphology of the microcapsules observed using confocal microscopy under different stirring rates. This was determined by calculating the ratio of intact microcapsules to all fragmented or damaged microcapsule structures in the image. The results are shown in Appendix A, where it can be seen that some of the microcapsules broke up after stirring, especially at 1000 rpm, where the fragmented structures significantly increased. However, for most of them, the complete microcapsule structure was maintained. This proves that the microcapsule structure is stable. It can be effectively loaded into the coating matrix.

### 3.2. Microcrack Self-Healing Performance Measurement

Self-healing microcapsules, as a type of exogenous self-healing solution; the presence of microcapsules themselves; and the addition of curing agents have the potential impact on the performance of the coating. Therefore, we tested the impact of different hardener dosages on the tensile strength of the coating under the same conditions. As shown in Figure 3a, the tensile strength of the coating significantly decreased with the increase in hardener dosage. To further evaluate the effect of adding microcapsules on tensile strength and repair efficiency under the same conditions, we chose 2 wt% as the fixed hardener addition amount and cut the coating samples into standard dumbbell shapes. By stretching the coating samples to 110% of their original length, we obtained damaged coatings with a fixed degree of damage. Then, the samples were placed at room temperature for 24 h to obtain the repaired coating samples. A schematic of the process is shown in Figure 3b. Figure 3c–e, respectively, represent the stress–strain curves of the coating before damage, after damage, and after repair. From these data, we found that the tensile strength of the coating significantly decreased after damage, but the tensile strength of the coating with added microcapsules significantly recovered after repair. The repair efficiency gradually increased with the increase in microcapsule addition, reaching a repair efficiency of up to 50.6%. The coating without microcapsules did not show significant changes. This is because during the stretching damage process, microcracks formed inside the coating due to stress, thereby reducing the overall tensile strength. In the coating with added microcapsules, the microcapsules near the cracks ruptured under stress during the damage process, subsequently releasing the healing agent to fill and repair the microcracks, thereby restoring the tensile strength to a certain extend, as presented in Figure 3g. However, when an excessive amount of microcapsules were present and unevenly distributed in the coating, areas with high microcapsule density and areas with sparse or no microcapsules could occur. This uneven distribution led to inconsistencies in the overall structure of the coating. Under mechanical stress or stretching, areas with a high concentration of microcapsules were more prone to damage. These areas could endure greater stress than other parts, leading to localized defects and, consequently, a reduction in the mechanical strength of the coating, as shown in Appendix A. Therefore, these data prove that this system can effectively heal minor damage inside waterborne polymer–cement coating, leading to the maintenance of mechanical properties in complex construction processes or application environments.

### 3.3. Filling Performance Measurement

In practical applications, waterproof coatings may suffer from large-scale macroscopic scratch damage due to external factors, such as being cut by sharp objects. This macroscopic damage also leads to the loss of the protective effect of the coating. Therefore, the macro-damage-healing effect of self-healing coatings was studied. A blade was used to make scratches 40~50 μm wide and 3 cm long on the surface of coatings with different amounts of microcapsule additions. We endeavored to create similar cracks on the surface of the coating through controlled methods such as adjusting the strength and cutting angle. We also set up parallel tests to reduce errors caused by differences in damage size. The scratch images at different times were obtained with a laser confocal microscope, and the depth of the scratches was measured. According to Figure 4a–c, over time, the cracks in the blank coating did not change significantly. However, the depth of the cracks in the damaged area of the coating with self-healing microcapsules continuously decreased. After 24 h of healing, when the microcapsule addition amount was 20 wt%, the depth of the crack decreased from 123.5 μm to 61.4 μm with a filling rate of 49.7%. As the amount of microcapsules added continued to increase, the filling rate of the cracks gradually increased, but the increase rate was relatively low. This is because the microcapsules in the coating, under blade cutting, ruptured along with the cracks in the coating. The healing agent, the EP stored in the microcapsules, automatically filled the cracks under its own fluidity and capillary action at the crack site. Importantly, this is not merely filling; in fact, the EP undergoes a curing reaction with the room-temperature curing agent AT30 dispersed in the coating, thereby effectively healing the crack.

To further explore the mechanical strength of the healing site, the coating samples with different amounts of microcapsules were directly cut off with a knife. The disconnected coatings were pieced together to heal for 24 h. The tensile strength of the coating after healing was tested using a universal testing machine, with the results presented in Appendix A. The results showed that the tensile strength of the completely disconnected coating reached up to 0.87 GPa after the healing. The coating sample containing 20 wt% self-healing microcapsules, after being cut and healed, could easily lift a 500 g, while the blank coating broke directly upon contact, as presented in Figure 4d and Appendix A. Therefore, the tensile strength of the coating with the self-healing microcapsules after the healing was much higher than that of the coatings without self-healing microcapsules.

### 3.4. Barrier Performance Measurement

The barrier performance of the coating was further evaluated by measuring |Z| at f = 0.01 Hz using a Bode plot [35,36]. The impact of adding self-healing microcapsules on the barrier performance of coatings was investigated by testing the electrochemical impedance spectra of the intact, damaged, and healing coatings. As shown in Figure 5a, after the addition of microcapsules, the electrochemical impedance of the coating in the low-frequency region significantly increased. The overlapping and interlocking arrangement of the EP microcapsule structures added to the coating seemed to greatly extend the penetration path length of water and electrolytes, thereby enhancing the coating’s barrier performance. However, with a microcapsule addition of more than 10 wt%, a slight decrease was observed. This could have been due to the uneven dispersion of too many microcapsules in the coating, creating structural defects in certain areas and slightly reducing the barrier performance of the intact coating. When the coating was damaged, the electrolyte directly contacted the substrate surface through the cracks, thus losing its barrier performance. As shown in Figure 5b, the average electrochemical impedance in the low-frequency range was about 10^3^. The electrochemical impedance spectrum of the damaged coating after 24 h healing is shown in Figure 5c, at which point the electrochemical impedance of the coating with added microcapsules in the 0.01 Hz region was higher than that before healing. Specifically, the value reached 85.7% when the microcapsule addition was about 20 wt%, as presented in Figure 5d. After the blank coating was damaged, the erosion derived from water and oxygen directly contacted the substrate at the damage site, thereby making the coating lose its protective effect. Conversely, the healing agent EP in the coating with added microcapsules was released from the microcapsules around the crack and solidified into a high-impedance EP protective film, thereby effectively restoring the barrier performance of the coating, as presented in Figure 5e,f. It is worth noting that the repair rate of the electrochemical barrier properties of the was is higher than the filling rate of 49.5%. Possibly, the electrochemical barrier properties of the cured EP were much higher than those of the coating. Therefore, even if the formed EP film was thinner than the coating, it provided excellent electrochemical barrier properties for that region, resulting in a higher repair efficiency. However, when too many microcapsules were added, after the healing agent fixed the coating, there may still have been some internal defects in the coating due to the uneven distribution of the microcapsules. Therefore, the presence of nonuniform phenomena during the healing process might have also adversely affected the repair efficiency [37]. To test the stability of the repaired coating, the repaired coating with a microcapsule addition of 20 wt% was immersed within a 3.5 wt% NaCl solution. The electrochemical barrier properties of the coating were tested after different immersion times, as shown in Appendix A. The healing coating exhibited excellent stability, which only showed a slight decrease in |Z| at f = 0.01 Hz after 150 h of immersion. The fitting model is shown in Appendix A. Rs is the solution resistance, Rc is the coating resistance, Rct is the charge transfer resistance, Cc is the capacitance, and Cdl is the double-layer capacitance. These data confirmed that the self-healing microcapsule system can heal large-scale damage to the coating, greatly restoring the structural strength and barrier performance of the damaged coating, to effectively extend the service life of the coating in complex environments.

## 4. Conclusions

In this study, a novel double-shelled self-healing microcapsule encapsulating EP was fabricated in waterborne polymer–cement coatings to endow self-healing properties and reduce cracking during coating drying, to effectively lower the coating application difficulty and lengthen its service life in complex environments. The microcapsules were prepared via in situ polymerization, including the initial embedding of the healing agent in a silica gel shell to form silica--gel-shelled microcapsules, and then depositing a metal-phenol network on the capsule surface via TA and Cu^2+^ chelation. The self-healing microcapsules had a uniform structure, with an average particle size of 64.06 μm and a wall thickness of 2–3 μm. Thus, it had an ultra-high sensitivity to damage and excellent healing with minor cracks produced during the internal drying of the coating. Specifically, the healing efficiency of minor cracks was around 50.8%, which effectively reduced cracking during the drying process of the coating. Moreover, for large–scale fully fractured damage, a crack–filling ratio of about 49.5% was achieved, with an electrochemical barrier healing efficiency of 85.7%. Even when completely severed, a tensile strength of around 0.87 Mpa could be attained after repair. In the future, we will further explore the application of these self-healing systems in other types of coatings, such as those used in the automotive or aerospace industries. These moisture-sensitive microcapsules have a wide range of potential applications in areas such as the medical field, pressure-sensitive materials, healthcare, and anticounterfeiting in printing.

## Figures and Tables

**Figure 1 polymers-16-00105-f001:**
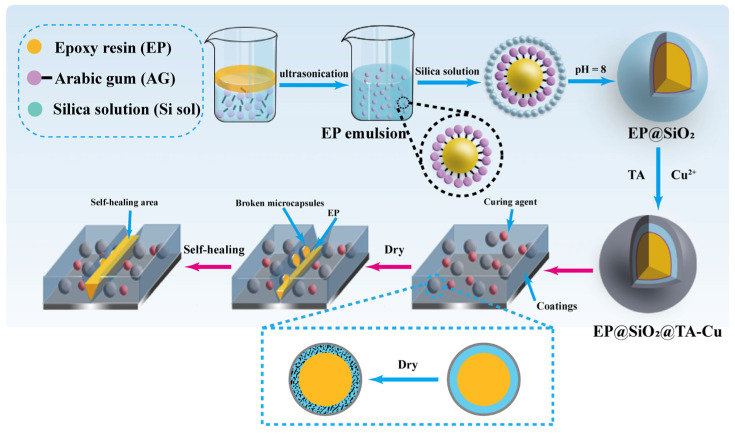
Diagram of self-healing microcapsules preparation and coating’s self-healing mechanism.

**Figure 2 polymers-16-00105-f002:**
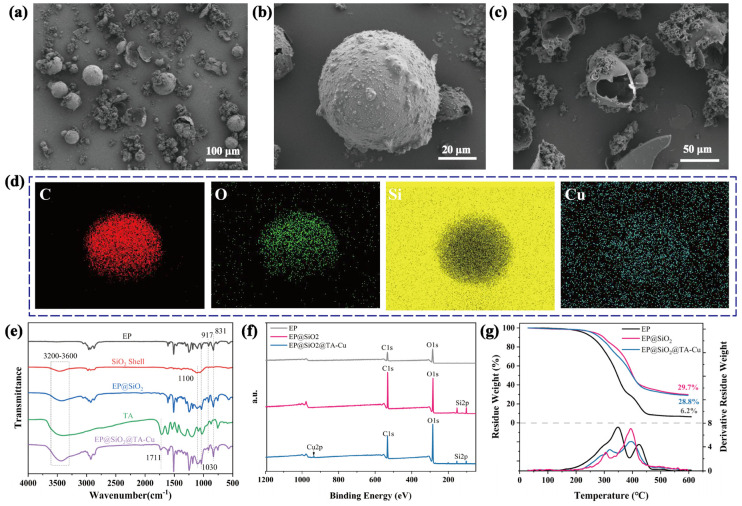
(**a**–**c**) SEM image of microcapsules; (**d**) elemental mapping images of microcapsules; (**e**) FTIR spectra of EP, SiO_2_ Shell, EP@SiO_2_, TA, and EP@SiO_2_@TA-Cu. (**f**) XPS analysis of EP, EP@SiO_2_, and EP@SiO_2_@TA-Cu; and (**g**) TGA thermograms of core and microcapsules.

**Figure 3 polymers-16-00105-f003:**
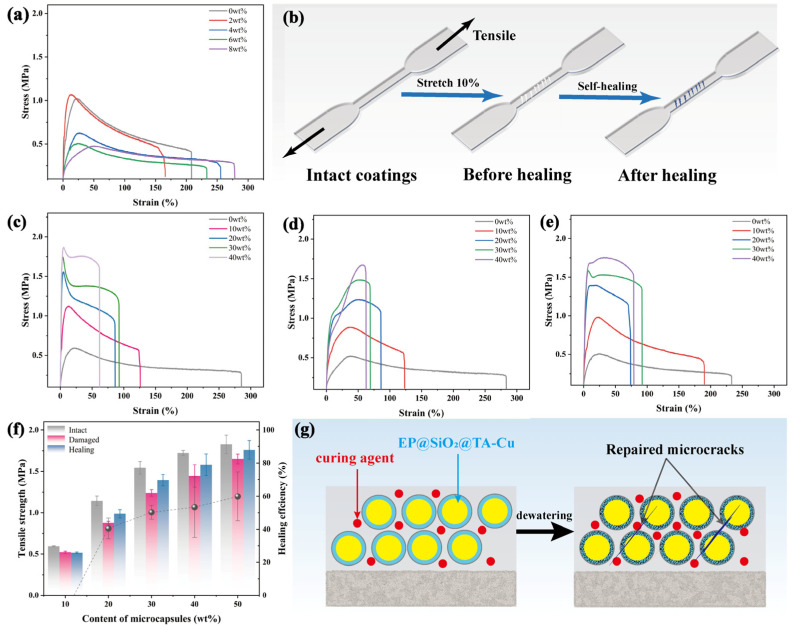
(**a**) Tensile stress–strain curve of the coating with different amounts of curing agent; (**b**) schematic of the simulated microcrack healing process sample preparation method; (**c**–**e**) tensile stress-strain curve of self-healing coatings with different microcapsules concentration: (**c**) intact coatings, (**d**) damaged coatings, (**e**) after healing coatings; (**f**) changes in tensile strength and self-healing rate for coatings with different microcapsules concentration; (**g**) schematic of microcrack healing during the coating drying and curing process.

**Figure 4 polymers-16-00105-f004:**
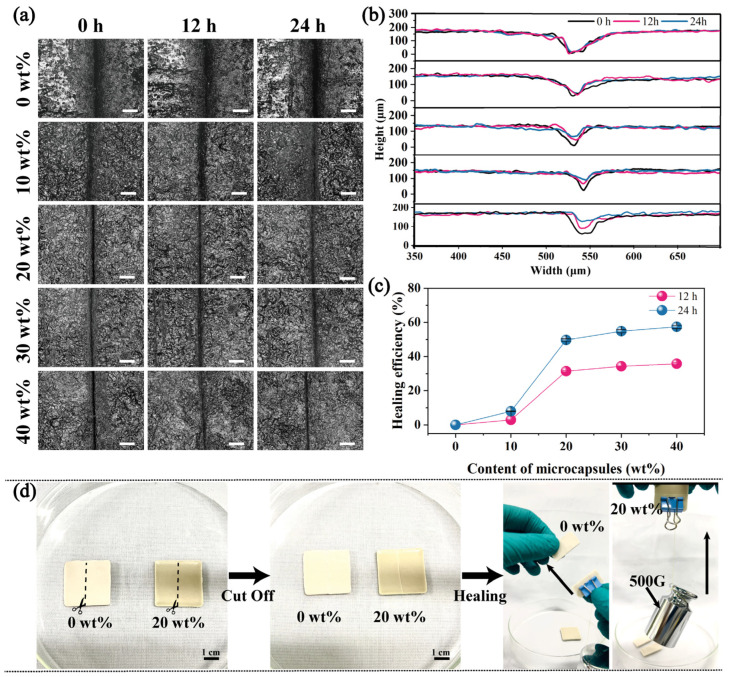
(**a**) POM images of self-healing coatings with different microcapsule concentrations 0 h, 12 h, and 24 h after scratching, respectively, scale bar: 100 μm; (**b**) scratch depth and healing rates of self-healing coatings with different microcapsule concentrations 0 h, 12 h, and 24 h after scratching; (**c**) healing rates of self-healing coatings with different microcapsules concentrations after 12 and 24 h. (**d**) Photo of the healing effect of the coating after complete cutting.

**Figure 5 polymers-16-00105-f005:**
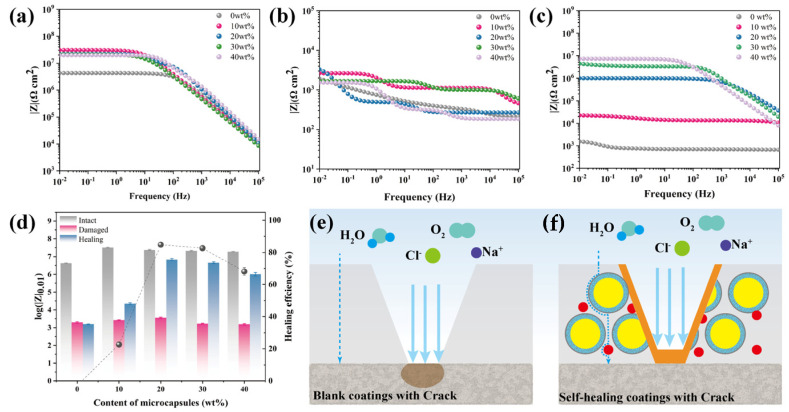
(**a**–**c**) Bode plots of self-healing coatings with different microcapsule concentrations immersed in 3.5 wt % NaCl solution: (**a**) Intact coatings, (**b**) damaged coatings, (**c**) after healing coatings; (**d**) impedance modulus |Z| measured at 0.01 Hz and healing efficiency during immersion in 3.5 wt % NaCl for self-healing coatings with different microcapsules concentration; (**e**,**f**) schematic of the barrier mechanism of blank coating and self-healing coating after damage.

## Data Availability

Data are contained within the article.

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
