# Peer review of "Dual-Shell Microcapsules for High-Response Efficiency Self-Healing of Multi-Scale Damage in Waterborne Polymer–Cement Coatings"

_polymers, 2023, doi:10.3390/polym16010105_

Round 1
Reviewer 1 Report
Comments and Suggestions for Authors
The manuscript ‘Dual-shell microcapsules for high-response efficiency self-healing of multi-scale damage in waterborne polymer-cement coatings’ presents the development of a silica gel shell and a tannin acid-cuprum dual-shell structured microcapsule to be utilized in polymer-cement coatings to enhance their self-healing ability. The authors have performed a lot of work and I believe that this manuscript can be published in Polymers; nevertheless, I have a few questions / comments that I would like the authors to respond to before publication.
It is not clear to me why self-healing can occur in larger-scale utilizing microcapsules than when intrinsic self-healing via reversible bonds is involved. Can the authors explain?
What is the error in the measurement of the size of the microcapsules? Is there accuracy on the second decimal point as the 64.06 mm indicate? I believe the authors should correct this.
The authors should pay attention on the numbering of the various parts of Figure 2 in the text. For example, Figure 2g is not SEM-EDS measurement nor Figure 2d is FTIR. Moreover, in the actual Figure 2g where the TGA measurements of the microcapsules are shown, I do not see any important difference in the weight loss curves and in the solid residue between EP@SiO2 and EP@SiO2@Ta-Cu. Is this expected / reasonable?
The authors claim that, under certain conditions of stirring, the microcapsules retain their structural integrity. However, looking at Figure S3, it seems that the majority of the capsules are broken. Am I wrong? The authors should explain what they mean by structural integrity and how this is concluded?
In the schematic of Figure 3g, the microcapsules seem to be very close and almost being in contact with each other. Is this correct? What would be the interparticle distance for a 10wt% dispersion of microcapsules of ~60mm. Does the schematic illustrate a different concentration?
What about the dispersion of the microcapsules in the polymer-cement matrix? How do the authors verify the good dispersion?
Figure 3 is not clear at all. I understand that Fig. 3a shows the stress-strain curves of coatings with 10wt% microcapsules and varying amount of curing agent. What is the difference in the systems of Figs 3c, 3d, 3e the behavior of which is examined for the different microcapsules concentration? What is the amount of curing agent in all of the systems of Figs 3c, 3d, 3e? Is this amount kept constant as the concentration of the microcapsules is increased and why? The authors should re-write this part since it is one of major importance for the manuscript.
How do the authors ensure that the cracks they are making on the surface of the coatings are similar in the different systems using manually a blade? I assume that the healing rate depends on the size of the crack therefore comparison of the healing process between similar cracks is critical. For example, the crack for the coating with no microcapsules (shown in Figure 4a) looks to me considerably larger than the ones with the microcapsules and actually it seems that the size of the crack becomes smaller as one goes from 0% to 10% to 20wt% and to 30% even at 0h. The authors should comment. Additionally, they can show how the healing process depends on the size of the crack.
Overall, I believe that this manuscript can be considered for publication in Polymers only after revision and only after the authors address the above-mentioned comments / questions.
Comments on the Quality of English LanguageI have no comments
Author Response
Dear Reviewer
Thank you very much for your valuable comments and suggestions, which were very helpful in the preparation of the revised manuscript. All the revisions, including our responses, are in the attached word document. In the document, the editor's and reviewer's comments are shown in deep red font and our responses are shown in the attachment.

Reviewer 2 Report
Comments and Suggestions for Authors
This manuscript presents an interesting approach for developing self-healing microcapsules for application in waterborne polymer-cement coatings. The dual-shell microcapsule design with a silica gel shell and metal-phenol network appears effective for embedding a healing agent that can be released upon coating damage to fill cracks. The microcapsules demonstrate suitable mechanical strength and stability for integration into coatings, while becoming more triggerable after drying. Experiments systematically vary microcapsule content and quantify self-healing ability on mechanical properties and barrier performance of coatings under both microcrack and macroscale damage conditions. Results are promising, with up to 50% strength recovery for microcracks and 86% crack depth reduction for cuts, also largely restoring coating protectiveness. Further optimization of microcapsule dispersion may improve performance. Overall the microencapsulation method and self-healing concept seem technically sound and well-executed, yielding coatings with enhanced durability.
- In the Introduction, provide more details on the specific chemical components and reactions involved in forming the dual-shell microcapsules to better set up the work.
- In Section 2.2, elaborate on the emulsification process and parameters (e.g. temperature, speed, time, surfactants). These details will help readers reproduce the microcapsule synthesis.
- in Figure 3 description, specify how microcapsule uneven distribution causes premature breakage.
- In Figure 4, add scale bar to microscopy images. And clarify if Figure S4 shows the same samples from Figure 4d.
- explain why increased microcapsule addition above 20 wt% does not proportionally increase filling rate further.
- For Figure 5 electrochemical impedance data, provide details on the spectra fitting model and parameters.
- In the Conclusion section, the authors may consider expanding on specific future work or applications of this microcapsule self-healing system.
- clarify in caption if size distribution is based on number frequency or volume frequency.
- Describe how the SEM, FTIR, XPS, TGA and other characterizations specifically help confirm the successful fabrication of the dual-shell microcapsules with silica gel core and metal-phenol network outer shell.
- Explain the testing standards and parameters for the mechanical testing, laser microscopy, and electrochemical impedance evaluations of the self-healing efficiency of the coatings.
- Provide more details on the polymer-cement coatings - their composition, chemistry, properties most relevant to self-healing ability. This gives context for why the microcapsules are effective.
- Elaborate on the proposed self-healing mechanism within the damaged coating based on the designed microcapsule structure and properties.
- Discuss any limitations of the current microcapsule system in terms of scale-up, optimization, stability, cost etc. and potential solutions.
- Compare performance of developed microcapsules versus existing state of the art systems from literature to benchmark advancements.
- Expand Figure 5 interpretation and significance of electrochemical impedance spectra results in showing self-healing ability.
- Perform a broader literature review of latest progress in self-healing coatings to identify open challenges that this work addresses and remaining issues for the field.
Minor editing of English language required.
Author Response
Dear reviewers
Thank you very much for your valuable comments and suggestions, which have been very helpful to us in writing the revised manuscript. All the revisions, including our responses, are in the attached word document. In the document, the editor's and reviewer's comments are shown in dark red font, and our responses are shown in black font. Please see the attachment.

Round 2
Reviewer 2 Report
Comments and Suggestions for Authors
Revised as required, worthy of publication